# Discretely-Refined Multi-view Clustering via Aligned Anchor Learning

Yuemeng Huang [1]  Huibing Wang [* 1]  Jinjia Peng [2]  Lu Jiang [1]  Qian Liu [1]  Zetian Mi [1]  Jiqing Zhang [1]

## Abstract

Anchor-based multi-view clustering has garnered wide attention for its ability to reduce the computational complexity of large-scale spectral clustering. However, existing methods mostly adopt a unidirectional optimization paradigm confined to sample-anchor bipartite graphs, treating the construction of the consensus graph and discrete clustering assignments as separate sub-problems to be solved independently. This weakens the information exchange between continuous representation and discrete structure, confining the optimization process to iterative updates within local modules. To address these limitations, we propose a Discretely-Refined Multi-view Clustering (DRMC) via Aligned Anchor Learning. Unlike approaches that directly perform fusion in the anchor space, our method starts from the anchor graph, elevates sample-anchor associations to sample-level similarity graph representations, and thereby enhances both within-cluster similarity and between-cluster separation. Furthermore, we design a discrete feedback module that jointly conducts spectral embedding learning and discrete label assignment by orthogonally aligning the continuous embedding matrix with the discrete indicator matrix. The resulting discrete partition is then fed back into the consensus graph construction, continuously refining the graph structure. Experiments on multiple benchmark datasets demonstrate that the proposed method exhibits significant advantages over existing state-of-the-art approaches.

[1]School of Information Science and Technology, Dalian Maritime University, Dalian, China [2]School of Cyber Security and Computer, Hebei University, Baoding, China. Correspondence to: Huibing Wang <huibing.wang@dlmu.edu.cn>.

*Proceedings of the 43rd International Conference on Machine Learning*, Seoul, South Korea. PMLR 306, 2026. Copyright 2026 by the author(s).

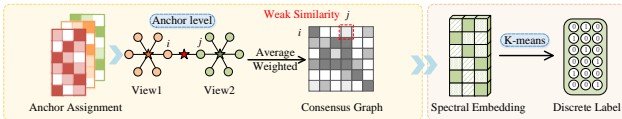

*(a)* Anchor-level multi-view clustering with post-spectral clustering

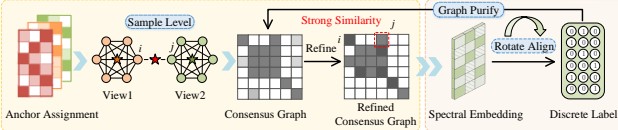

*(b)* Sample-level multi-view clustering with graph purification

*Figure 1.* Comparison between existing AMVC methods and our proposed approach. (a) Existing methods employ anchor-level alignment and obtain discrete labels through separate spectral clustering followed by post-processing k-means. (b) DRMC achieves fine-grained sample-level alignment and introduces graph purification to learn discrete labels.

## 1. Introduction

Multi-view data originates from describing the same object from different dimensions or modalities (Wang et al., 2026; 2023; Sun et al., 2024). For example, the same news event can be presented in various forms such as images, text, and audio, each constituting a distinct view (Yuan et al., 2025; 2024; Sun et al., 2025). This kind of data naturally contains rich complementary information, which can more comprehensively reveal the internal structure and semantic characteristics of the object. Multi-view clustering (MVC) integrates effective information from different views to uncover the underlying cluster structure in the data (Zhou et al., 2024; Yan et al., 2023; Xu et al., 2022), and is therefore widely used in fields such as medical image analysis and multimedia retrieval.

However, existing graph-based multi-view clustering methods usually rely on sample similarity graphs, which leads to increasing computational and storage costs as the number of samples grows. To improve graph construction and representation learning, anchor-based MVC methods (AMVC) (Cui et al., 2025; Shang et al., 2025; Wang et al., 2025) introduce representative anchors to model the associations between samples and anchors.

Early AMVC methods often employed heuristic sampling strategies, separating anchor selection from subsequent

graph learning, which easily led to suboptimal solutions. To address this, researchers proposed unified optimization frameworks that jointly optimize anchor learning and graph construction. For example, both (Wang et al., 2021) and (Liu et al., 2022) adopted this idea, with the latter further introducing graph connectivity constraints to directly output discrete labels, eliminating the need for additional discretization steps. Building on this, effectively integrating multi-view information to learn a high-quality consensus graph has become crucial. (Fang et al., 2023) proposed adaptively fusing multi-view anchor graphs while imposing Laplacian rank constraints to learn a consensus graph with clear cluster structure. (Shu et al., 2022) utilized the tensor Schatten $p$-norm to explore the complementary and low-rank properties among multi-view anchor graphs, learning a consensus graph in a self-weighted manner. These methods have significantly improved the quality of the consensus graph.

To adapt to the diversity and complexity of data in real-world scenarios, AMVC methods have continuously evolved. (Qu et al., 2025) first proposed an anchor-guided incremental alignment framework for incremental data where samples and features are not aligned. (Zhang et al., 2024) focused on continuous learning scenarios, reusing historical anchor information through consistency regularization to guide the learning of new data. To improve the flexibility of the algorithm, (Zhang et al., 2023) allowed different views to use different numbers of anchors and adaptively fused multi-size anchor graphs, avoiding manual parameter tuning of the number of anchors. (Zhao et al., 2025) combines unsupervised regression with anchor graph learning, eliminating the complex graph alignment steps and opening up a new path in this direction.

Despite significant progress, existing methods still face notable challenges, as depicted in Figure 1(a). They typically adopt a segmented optimization approach, where cross-view alignment is often performed at the level of sample-anchor bipartite graphs rather than directly on more discriminative sample-level similarity structures. This leads to inadequate consensus representation. Moreover, graph learning and discrete assignment follow a unidirectional process, preventing clustering results from providing feedback to refine the graph structure, thereby limiting clustering performance and compromising stability.

To address the aforementioned issues, we introduce a Discretely-Refined Multi-view Clustering method (DRMC) via Aligned Anchor Learning, as shown in Figure 2. First, the framework introduces a sample-level view alignment mechanism. Unlike directly fusing at the sample-anchor bipartite graph level, this method elevates the anchor-based similarity relations into a sample-level similarity graph, thereby aligning structural information across different

*Table 1.* Notations and Descriptions of The Main Formulas

| Notations | Descriptions |
|---|---|
| $n,k,V$ | Number of samples, clusters and views |
| $m$ | Number of anchors |
| $d_v$ | Feature dimension of the $v$-th view |
| $\alpha$ | View weight vector |
| $\mathbf{X}^{(v)} \in \mathbb{R}^{n \times d_v}$ | Data matrix for the $v$-th view |
| $\mathbf{A}^{(v)} \in \mathbb{R}^{n \times m}$ | Anchor assignment matrix for the $v$-th view |
| $A_{ij}^{(v)}$ | Affiliation strength $x_i^{(v)}$ and $j$-th anchor |
| $\mathbf{U}^{(v)} \in \mathbb{R}^{m \times d_v}$ | Anchor feature matrix for the $v$-th view |
| $\mathbf{W}^{(v)} \in \mathbb{R}^{n \times n}$ | Similarity matrix for the $v$-th view |
| $\overline{\mathbf{W}} \in \mathbb{R}^{n \times n}$ | Consensus similarity matrix |
| $\mathbf{F} \in \mathbb{R}^{n \times k}$ | Spectral embedding matrix |
| $\mathbf{Y} \in \mathbb{R}^{n \times k}$ | Discrete label matrix |
| $\mathbf{R} \in \mathbb{R}^{k \times k}$ | Rotation matrix |

views at the sample level. On this basis, an adaptive weighting strategy is applied to fuse the views, which enhances the contribution of high-quality views and suppresses the interference from noisy views. More critically, the method abandons the conventional two-stage approach that separates spectral embedding and discretization. Instead, it introduces orthogonal alignment to establish a direct geometric correspondence between the continuous spectral embedding space and the discrete clustering indicator space. On this basis, the current discrete labels are leveraged to dynamically purify the consensus graph, allowing the clustering information to be fed back into the graph learning process. This mutual correction mechanism between the graph structure and the discrete labels forms a bidirectional optimization loop, significantly enhancing the model's robustness to noise. Compared with existing algorithms, the main contributions of this paper are summarized as follows:

- We design a sample-level view alignment and fusion mechanism, which elevates the anchor graphs from each view into complete sample-to-sample similarity matrices, thereby aligning cross-view structural information at the sample level.

- We construct a novel bidirectional feedback module. It links continuous spectral embedding with discrete cluster indicators, and utilizes the labels to dynamically purify the consensus graph, forming a feedback loop between label assignment and graph purification.

- We employ an adaptive weighting strategy to learn a more discriminative consensus graph. Extensive experiments on multiple standard datasets demonstrate that our method achieves good clustering results on most datasets compared to existing state-of-the-art methods.

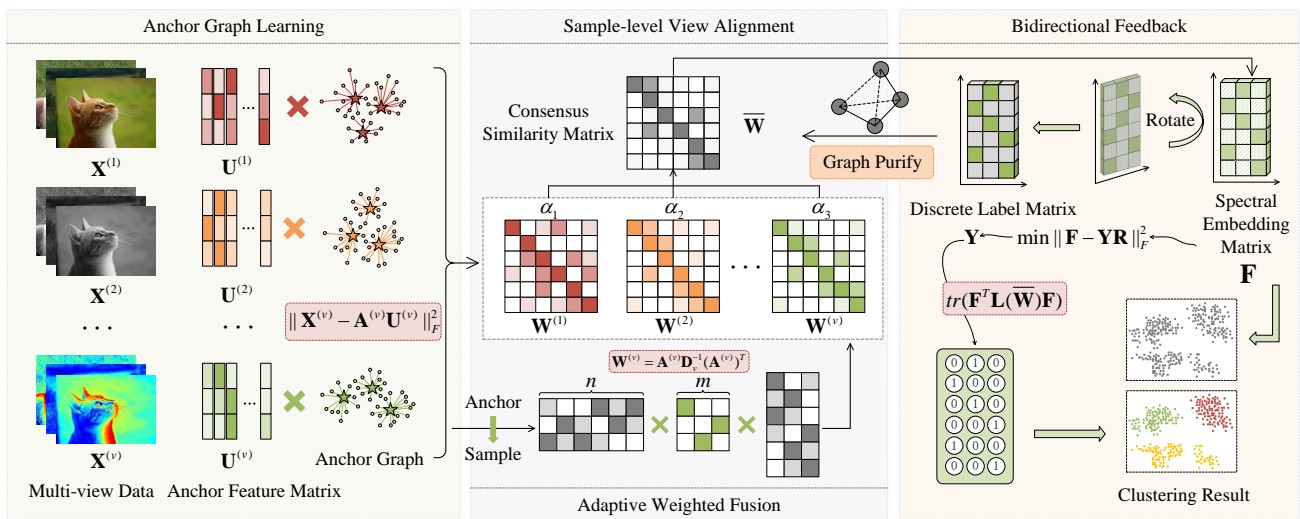

*Figure 2.* The framework of DRMC. First, matrix factorization is employed to learn the assignment relationship between samples and anchors as well as the anchor feature representation. Second, based on fine-grained sample-level similarity, the graph structure of each view is constructed and fused via adaptive weighted fusion to generate the consensus graph. Finally, a discrete feedback module is introduced to achieve collaborative optimization between the graph structure and cluster labels, where the discrete labels are used for graph purification to further enhance the quality of the consensus graph.

## 2. Methodology

This section first summarizes the key notations used in this paper in Table1, followed by an elaboration of the motivation behind the proposed method, its specific details, and an analysis of its computational complexity.

### 2.1. Formulation

***Anchor Graph Learning:*** To learn discriminative anchor representations from each view, we employ matrix factorization to approximate the original data matrix as the product of an anchor assignment matrix and a feature matrix. To enhance the robustness of the representations, a sparse regularization term is introduced into the objective function, while non-negativity and row-normalization constraints are imposed on the assignment matrix, ensuring a clear probabilistic interpretation. This can be formulated as:

$$\min_{\mathbf{A}^{(v)}, \mathbf{U}^{(v)}} \left\| \mathbf{X}^{(v)} - \mathbf{A}^{(v)}\mathbf{U}^{(v)} \right\|_F^2 + \rho\,\Omega\left(\mathbf{A}^{(v)}\right) \quad (1)$$
$$\text{s.t. } \mathbf{A}^{(v)} \geq 0, \mathbf{U}^{(v)} \geq 0, \mathbf{A}^{(v)}\mathbf{1} = \mathbf{1}.$$

Each row vector of the anchor assignment matrix $\mathbf{A}^{(v)}$ represents the allocation probability of a sample across the anchors.

***Sample-level view alignment and adaptive weighted fusion:*** To achieve fine-grained alignment across view structures at the sample level, we upgrade it to a complete inter-sample similarity matrix representation based on the learned anchor assignment matrix $\mathbf{A}^{(v)}$ for each view. Specifically, for the

$v$-th view:

$$\mathbf{W}^{(v)} = \mathbf{A}^{(v)}\mathbf{D}_v^{-1}(\mathbf{A}^{(v)})^T \quad (2)$$

where $\mathbf{D}_v = diag((\mathbf{A}^{(v)})^T\mathbf{1})$ is the anchor degree matrix. This process transforms the local association between samples and anchors into global sample similarity, establishing a comprehensive sample-level representation within each view. Based on this, to achieve sample-level structural alignment across views, we fuse the sample similarity matrices of each view into a unified consensus graph $\overline{\mathbf{W}}$. Simultaneously, we introduce a view weight vector $\alpha = [\alpha_1, ..., \alpha_V]^T$ and achieve adaptive fine-grained fusion by optimizing the following objectives:

$$\min_{\alpha, \mathbf{W}} \sum_{v=1}^{V} \alpha_v \left\| \mathbf{W}^{(v)} - \mathbf{W} \right\|_F^2 \quad \text{s.t. } \alpha \geq 0, \mathbf{1}^T\boldsymbol{\alpha} = 1. \quad (3)$$

By fusing at the sample-level similarity matrix level, the model can more accurately capture structural consistency, thereby learning a more robust consensus representation.

***Bidirectional Feedback:*** Based on the obtained consensus graph, spectral embedding learning and discretization are further performed. Traditional multi-view clustering methods often treat spectral embedding and discretization as two separate stages, which may lead to inconsistency between the continuous representations and the final discrete cluster assignments. To address this, we introduce a discrete consistency constraint into a unified optimization framework and establish a bidirectional synergy between the discretization process and graph structure learning. First, we construct the normalized Laplacian matrix $\mathbf{L}_{(\overline{\mathbf{W}})} = \mathbf{I} - \mathbf{D}^{-\frac{1}{2}}\mathbf{W}\mathbf{D}^{-\frac{1}{2}}$, where $\mathbf{D} = \mathrm{diag}(\overline{\mathbf{W}}\mathbf{1})$. By

performing eigen-decomposition on $\mathbf{L}_{(\overline{\mathbf{W}})}$ and selecting the eigenvectors corresponding to its first k smallest eigenvalues, we obtain the continuous spectral embedding matrix $\mathbf{F} \in \mathbb{R}^{n \times k}$, i.e., $\text{tr}(\mathbf{F}^T \mathbf{L}_{(\overline{\mathbf{W}})} \mathbf{F})$. Simultaneously, a discrete consistency constraint is introduced to minimize the distance between the continuous embedding $\mathbf{F}$ and the discrete labels $\mathbf{Y}$ under an orthogonal transformation, thereby promoting the alignment of the spectral embedding with the discrete space:

$$\min_{\mathbf{Y}, \mathbf{R}} \|\mathbf{F} - \mathbf{YR}\|_F^2 \quad s.t. \ \mathbf{Y} \in \{0,1\}^{n \times k}, \mathbf{Y1} = \mathbf{1}, \mathbf{R}^T \mathbf{R} = \mathbf{I}. \tag{4}$$

To further suppress potential noisy edges in the consensus graph and enhance intra-cluster consistency, we further propose a discrete consistency-driven graph calibration mechanism. Based on the current discrete partitions, the consensus graph structure is updated.

Integrating the above modules, the overall objective function of the model can be written as follows:

$$\begin{aligned} \min \quad & \sum_{v=1}^{V} \left\|\mathbf{X}^{(v)} - \mathbf{A}^{(v)}\mathbf{U}^{(v)}\right\|_F^2 + \rho \sum_{v=1}^{V} \Omega\left(\mathbf{A}^{(v)}\right) \\ & + \lambda \left( \sum_{v=1}^{V} \alpha_v \left\|\mathbf{W}^{(v)} - \overline{\mathbf{W}}\right\|_F^2 + \text{tr}\left(\mathbf{F}^T \mathbf{L}_{(\overline{\mathbf{W}})} \mathbf{F}\right) \right. \\ & \left. + \|\mathbf{F} - \mathbf{YR}\|_F^2 + \mu \left\|\left(\mathbf{11}^T - \mathbf{S}\right) \odot \overline{\mathbf{W}}\right\|_F^2 \right) \\ s.t. \quad & \mathbf{S} = \mathbf{YY}^T, \mathbf{A}^{(v)} \ge 0, \mathbf{U}^{(v)} \ge 0, \mathbf{A}^{(v)}\mathbf{1} = \mathbf{1}, \\ & \boldsymbol{\alpha} \ge 0, \quad \mathbf{1}^T\boldsymbol{\alpha} = 1, \mathbf{Y} \in \{0,1\}^{n \times k}, \mathbf{Y1} = \mathbf{1}, \mathbf{R}^T\mathbf{R} = \mathbf{I}. \end{aligned} \tag{5}$$

## 3. Optimization

To address the aforementioned multi-variable optimization problem, we employ an alternating optimization strategy. This approach decomposes the original problem into several subproblems, where each variable is updated while keeping the others fixed, thereby transforming the complex problem into a series of efficiently solvable subproblems.

**1) $\mathbf{U}^{(v)}$ Sub-Step:** When other variables are fixed, the subproblem w.r.t. $\mathbf{U}^{(v)}$ is

$$\min_{\mathbf{U}^{(v)}} \left\|\mathbf{X}^{(v)} - \mathbf{A}^{(v)}\mathbf{U}^{(v)}\right\|_F^2 \quad s.t. \ \mathbf{U}^{(v)} \ge 0. \tag{6}$$

To ensure non-negativity, we follow the multiplicative update rule and introduce a small regularization term for numerical stability, yielding:

$$\mathbf{U}^{(v)} \leftarrow \mathbf{U}^{(v)} \odot \frac{\mathbf{A}^{(v)^T}\mathbf{X}^{(v)}}{\mathbf{A}^{(v)^T}\mathbf{A}^{(v)}\mathbf{U}^{(v)} + \varepsilon} \tag{7}$$

**2) $\mathbf{A}^{(v)}$ Sub-Step:** When other variables are fixed, the subproblem w.r.t. $\mathbf{A}^{(v)}$ is

$$\begin{aligned} \min_{\mathbf{A}^{(v)}} \left\|\mathbf{X}^{(v)} - \mathbf{A}^{(v)}\mathbf{U}^{(v)}\right\|_F^2 + \rho\Omega\left(\mathbf{A}^{(v)}\right) \\ s.t. \ \mathbf{A}^{(v)} \ge 0, \mathbf{A}^{(v)}\mathbf{1} = \mathbf{1}. \end{aligned} \tag{8}$$

To enhance the discriminative power of anchor assignments, we define a sparse regularization term as $\Omega(\mathbf{A}^{(v)}) = \sum_{i=1}^{n} \sum_{j=1}^{m} \frac{\mathbf{A}_{ij}^{(v)2}}{\mathbf{A}_{ij}^{(v)} + \varepsilon}$. To facilitate the multiplicative update process, we define a reweighting matrix as $\mathbf{W}_{ij}^{(v,t)} = \frac{\mathbf{A}_{ij}^{(v,t-1)}}{\left(\mathbf{A}_{ij}^{(v,t-1)} + \varepsilon\right)^2}$. Using the multiplicative update rule, we obtain:

$$\mathbf{A}^{(v)} \leftarrow \mathbf{A}^{(v)} \odot \frac{\mathbf{X}^{(v)}(\mathbf{U}^{(v)})^T}{\mathbf{A}^{(v)}\mathbf{U}^{(v)}(\mathbf{U}^{(v)})^T + \rho\mathbf{W}^{(v,t)} + \varepsilon} \tag{9}$$

**3) $\mathbf{W}^{(v)}$ Construction:** $\mathbf{W}^{(v)}$ is not treated as an independent optimization variable; Instead, the initial similarity matrix is computed via the anchor assignment matrix $\mathbf{A}^{(v)}$:

$$\mathbf{W}^{(v)} = \mathbf{A}^{(v)}\mathbf{D}_a^{-1}(\mathbf{A}^{(v)})^T \tag{10}$$

where $\mathbf{D}_a = diag\left(\sum_{i=1}^{n} \mathbf{A}_{ij}^{(v)}\right)$ is the anchor degree matrix. To enhance discriminative power and reduce computational complexity, we retain only the top-$q$ connections with the largest weights for each row of $\mathbf{W}^{(v)}$

$$\mathbf{W}_{ij}^{(v)} \leftarrow \begin{cases} \mathbf{W}_{ij}^{(v)}, j \in \mathcal{N}_q(i) \\ 0, otherwise \end{cases} \tag{11}$$

where $\mathcal{N}_q(i)$ denotes the set of the $q$ nearest neighbors with the highest similarity to sample $i$. Finally, by applying symmetric normalization to the graph, we obtain:

$$\mathbf{D}_v = diag\left(\sum_{j=1}^{n} \mathbf{W}_{ij}^{(v)}\right), \mathbf{W}^{(v)} \leftarrow \mathbf{D}_v^{-\frac{1}{2}}\mathbf{W}^{(v)}\mathbf{D}_v^{-\frac{1}{2}} \tag{12}$$

**4) $\alpha$ Sub-Step:** When other variables are fixed, the view weight $\alpha$ is used to measure the reliability of each view-specific graph. A view-specific graph closer to the consensus graph is considered more reliable and should contribute more to the final fusion. Therefore, we adopt a simple distance-inverse weighting strategy:

$$d_v = \|\mathbf{W}^{(v)} - \overline{\mathbf{W}}\|_F + \varepsilon \tag{13}$$

where $\epsilon$ is a small constant to avoid numerical instability. The view weight is then updated by

$$\alpha_v = \frac{d_v^{-1}}{\sum_{u=1}^{V} d_u^{-1}}, \quad v = 1, \ldots, V \tag{14}$$

This strategy assigns larger weights to views that are more consistent with the current consensus graph while maintaining $\alpha_v \geq 0$ and $\mathbf{1}^T \alpha = 1$.

**5) $\overline{\mathbf{W}}$ Sub-Step:** When other variables are fixed, the subproblem w.r.t. $\overline{\mathbf{W}}$ is

$$\min_{\overline{\mathbf{W}}} \left\| \mathbf{C} - \overline{\mathbf{W}} \right\|_F^2 + \mu \left\| \left( \mathbf{1}\mathbf{1}^T - \mathbf{S} \right) \odot \overline{\mathbf{W}} \right\|_F^2 \qquad (15)$$

where $\mathbf{C} = \sum_{v=1}^{V} \alpha_v \mathbf{W}^{(v)}$ indicates the weighted aggregation of view-specific graphs. $\mathbf{S} = \mathbf{Y}\mathbf{Y}^T$, where $S_{ij} = 1$ indicates that samples $i$ and $j$ are assigned to the same cluster, and $S_{ij} = 0$ otherwise. Since the above problem is separable with respect to each element of $\overline{\mathbf{W}}$, the element-wise solution is $\overline{W_{ij}} = \frac{C_{ij}}{1+\mu(1-S_{ij})}$. Therefore, we can obtain:

$$\overline{\mathbf{W}} = \mathbf{S} \odot \mathbf{C} + \gamma \left( \mathbf{1}\mathbf{1}^T - \mathbf{S} \right) \odot \mathbf{C} \qquad (16)$$

where $\gamma = \frac{1}{1+\mu}$. Therefore, intra-cluster edges are preserved, while inter-cluster edges are adaptively suppressed. After this update, $\overline{\mathbf{W}}$ is symmetrized and normalized to ensure a valid similarity graph.

**6) R Sub-Step:** When other variables are fixed, the subproblem w.r.t. $\mathbf{R}$ is The spectral embedding matrix $\mathbf{R}$ is obtained by solving the following optimization problem:

$$\min_{\mathbf{R}} \|\mathbf{F} - \mathbf{Y}\mathbf{R}\|_F^2 \quad \text{s.t. } \mathbf{R}^T\mathbf{R} = \mathbf{I}. \qquad (17)$$

Since this is an orthogonal Procrustes problem, a closed-form solution can be obtained via singular value decomposition. Let $\mathbf{M} = \mathbf{Y}^T\mathbf{F} \in \mathbb{R}^{k \times k}$. Performing singular value decomposition on $\mathbf{M}$ yields $\mathbf{M} = \mathbf{U}\boldsymbol{\Sigma}\mathbf{V}^T$, where $\mathbf{U}$ and $\mathbf{V}$ are orthogonal matrices, and $\boldsymbol{\Sigma}$ is a diagonal matrix. Thus, the optimal rotation matrix is given by $\mathbf{R} = \mathbf{U}\mathbf{V}^\mathbf{T}$.

**7) Y Sub-Step:** When other variables are fixed, the subproblem w.r.t. $\mathbf{Y}$ is

$$\min_{\mathbf{Y}} \|\mathbf{F} - \mathbf{Y}\mathbf{R}\|_F^2 . \qquad (18)$$

The above optimization problem is equivalent to

$$\max_{\mathbf{Y}} \text{tr} \left( \mathbf{F}^T\mathbf{Y}\mathbf{R} \right) = \max_{\mathbf{Y}} \sum_{i=1}^{n} \sum_{j=1}^{k} \left( \mathbf{F}\mathbf{R}^T \right)_{ij} \mathbf{Y}_{ij}$$
$$s.t. \sum_{j=1}^{k} \mathbf{Y}_{ij} = 1, \forall i \qquad (19)$$

Since the rows of $\mathbf{Y}$ are independent, each sample $i$ can be solved independently:

$$\max_{j \in \{1,...,k\}} \left( \mathbf{F}\mathbf{R}^T \right)_{ij}, \quad \forall i = 1,...,n \qquad (20)$$

---

**Algorithm 1 DRMC** algorithm

**Input**: Multi-view data $\{\mathbf{X}^{(v)}\}_{v=1}^V$, cluster number $k$, anchor number $m$, $\lambda$, $\rho$, $\mu$, max_$iter$, set $\alpha$ to $\frac{1}{V}$.

**Output**: Discrete label matrix $\mathbf{Y}$ and clustering results.

1: Initialize: $\mathbf{A}^{(v)}, \mathbf{U}^{(v)}, \overline{\mathbf{W}}, \mathbf{F}, \mathbf{Y}, \mathbf{R}$
2: **while** not converged **do**
3:     Update variable $\mathbf{U}^{(v)}$ using Eq. (7);
4:     Update variable $\mathbf{A}^{(v)}$ using Eq. (9);
5:     Construct the view-specific graph $\mathbf{W}^{(v)}$ based on $\mathbf{A}^{(v)}$ using Eq. (10)-(12);
6:     Update variable $\alpha$ using Eq. (14);
7:     Construct $\mathbf{S} = \mathbf{Y}\mathbf{Y}^T$ using the current discrete label matrix $\mathbf{Y}$;
8:     Update variable $\overline{\mathbf{W}}$ using the label-guided graph purification rule in Eq. (16);
9:     Obtain $\mathbf{F}$ by performing eigen decomposition on the Laplacian matrix of $\overline{\mathbf{W}}$.
10:    Update variable $\mathbf{R}$ using Eq. (17);
11:    Update variable $\mathbf{Y}$ using Eq. (21);
12: **end while**

---

Therefore, the optimal discrete label assignment can be obtained as:

$$\mathbf{Y}_{ij} = \begin{cases} 1, & j = \arg\max_{l} \left( \mathbf{F}\mathbf{R}^T \right)_{il} \\ 0, & \text{otherwise} \end{cases} \qquad (21)$$

It should be noted that although $\mathbf{S} = \mathbf{Y}\mathbf{Y}^T$ in graph purification is related to the discrete label $\mathbf{Y}$, this paper does not directly incorporate this term into the subproblem for solving $\mathbf{Y}$. Specifically, $\mathbf{Y}$ is first updated via the discrete consistency constraint; then, $\mathbf{S}$ is constructed based on the current $\mathbf{Y}$ and used as discrete feedback information for the subsequent update of $\overline{\mathbf{W}}$. In this way, the simplicity of the discrete label update is preserved while leveraging the label information to back-calibrate the consensus graph structure.

*Table 2.* Details for Multi-view Datasets

| Dataset | Sample | Cluster | View | Feature Dimension |
|---|---|---|---|---|
| 3sources | 169 | 6 | 3 | 3560/3631/3068 |
| MSRC_v1 | 210 | 7 | 5 | 24/576/512/256/254 |
| BBC4 | 685 | 5 | 4 | 4659/4633/4665/4680 |
| BDGP_fea | 2500 | 5 | 3 | 100/500/250 |
| Wiki | 2866 | 6 | 2 | 128/10 |
| Caltech101-20 | 2386 | 20 | 6 | 48/40/254/1984/512/928 |
| Caltech101-all | 9144 | 102 | 6 | 48/40/254/1958/512/928 |
| AwA | 30475 | 50 | 6 | 2688/2000/252/2000/.../2000 |

## 4. Complexity Analysis

In this section, we provide a detailed analysis of the time and space complexity of the proposed algorithm.

**Time complexity:** The time computational cost of the algorithm mainly comes from the alternating optimization pro-

cess across different modules. The time complexity of updating $\mathbf{U}^{(v)}$ is $\mathcal{O}(nmd_v + nm^2 + m^2d_v)$. The time complexity of updating $\mathbf{A}^{(v)}$ is also $\mathcal{O}(nmd_v + m^2d_v + nm^2)$. The time complexity of updating $\mathbf{W}^{(v)}$ is $\mathcal{O}(n^2m + n^2\log n)$. The cost of updating $\alpha$ and $\overline{\mathbf{W}}$ are $\mathcal{O}(n^2)$. Computing the spectral embedding $\mathbf{F}$ takes $\mathcal{O}(n^2k)$. The steps of discretization and graph purification together require $\mathcal{O}(nk^2 + k^3 + n^2)$. Since in practice $d_v, k, m \ll n$, the overall time complexity of the algorithm is $\mathcal{O}(n^2)$.

**Space complexity:** The main data stored by the algorithm and their space requirements are as follows: the original data occupies $\mathcal{O}(nd_v)$, the anchor assignment matrix requires $\mathcal{O}(nm)$, the anchor feature matrix takes $\mathcal{O}(md_v)$, each view-specific similarity graph needs $\mathcal{O}(n^2)$, both the consensus graph and its corresponding Laplacian matrix require $\mathcal{O}(n^2)$, the spectral embedding matrix uses $\mathcal{O}(nk)$, and the discrete label matrix occupies $\mathcal{O}(nk)$. Therefore, the overall space complexity of the algorithm is $\mathcal{O}(n^2)$.

# 5. Experiments

To evaluate the effectiveness of the proposed method, we compared its performance and efficiency with 11 baseline methods on 8 datasets.

## 5.1. Experimental Setting

**Datasets:** We conducted comparative experiments on eight datasets: 3sources, MSRC_v1, BBC4, BDGP, Wiki, Caltech101-20, Caltech101-all and AwA, along with detailed information Table2.

**Comparison Methods:** To validate the effectiveness of the proposed method, we compare it with eleven advanced AMVC methods, including SMVSC(Sun et al., 2021), OPMC(Liu et al., 2021), OMSC(Chen et al., 2022), FPMVS(Wang et al., 2021), EOMSC(Liu et al., 2022), AWMVC(Wan et al., 2023), FDAGF(Zhang et al., 2023), OMVCDR(Wan et al., 2024), RCAGL(Liu et al., 2024), ALPC(Chen et al., 2025), and TLRLF4MVC(Long et al., 2025). SMVSC and OMSC focus on the joint optimization of anchor learning and graph construction; OPMC, FPMVS, EOMSC, AWMVC, FDAGF, and OMVCDR emphasize efficient multi-view fusion mechanisms and one-step clustering frameworks; RCAGL and TLRLF4MVC aim to enhance the robustness of representation learning, with TLRLF4MVC introducing a tensor low-frequency component (TLFC) operator combined with low-rank constraints to learn smooth and structurally consistent embedding representations.

**Parameter Setup:** To ensure a fair comparison, the best recorded performance metrics are reported for all baseline methods. The parameters of our proposed model are set as follows: $\lambda$ is selected from the range $[1e^0, 1e^7]$, $\rho$ is chosen from $[1e^{-7}, 1e^{-1}]$, and the number of anchors is adjusted within $\{3k, 4k, 5k\}$, where $k$ denotes the number of clusters.

**Evaluation Metrics:** We employ four evaluation metrics: ACC, NMI, Purity, and F-score. For all these metrics, higher values correspond to better model performance.

## 5.2. Experimental Results and Analysis

Table3 presents a comparison of the proposed method and the 11 baseline methods across the four evaluation metrics. In the table, the best results are highlighted in red, and the second-best results are marked in blue. Based on the data in Table3, the following conclusions can be drawn: (1)Most existing methods typically rely on algorithms such as k-means for one-time discretization after obtaining spectral embeddings, making it difficult to further correct cross-cluster noisy edges once the consensus graph is constructed. In contrast, the DRMC method introduces a graph purification mechanism based on discrete feedback, which continuously optimizes graph structure quality and thereby significantly enhances clustering performance. (2) Our method achieves the competitive performance and obtains the best results on many datasets. In particular, it outperforms the second-best result by 9.47 on the 3sources dataset, which strongly demonstrates the effectiveness of sample-level view alignment in learning consistent cross-view structures. (3)Among the comparative methods, AWMVC, FDAGF, and RCAGL also deliver strong results on certain datasets. However, their performance tends to be unstable. For example, AWMVC performs well on 3sources but shows only moderate performance on Caltech101-20. In contrast, our method achieves more stable and leading results across most datasets. (4) Although some metrics do not achieve the highest values on certain datasets, our method still maintains a clear advantage in terms of ACC and Purity.

## 5.3. Parameter Sensitivity and Convergence Analysis

We further examined the dependence of model on the $\lambda$ and $\rho$ through a sensitivity analysis. As shown in Figure 3, the evaluation metrics show little fluctuation across parameter adjustments on all three datasets. This observation indicates that the model maintains stable performance over a broad parameter range, exhibiting good robustness.

To investigate the impact of the number of anchors on clustering performance, we conducted a sensitivity experiment with respect to the anchor quantity. As shown in Figure 4, the number of anchors is varied from $2k$ to $5k$, and the performance is evaluated using ACC, NMI, Purity, and F-score. The results show that both datasets achieved optimal performance when the number of anchors is $5k$, while some metrics significantly declined with fewer anchors. This indicates that an insufficient number of anchors leads to information loss, thereby impairing clustering performance.

*Table 3.* Comparison with multi-view clustering methods.

| Datasets | SMVSC | OPMC | OMSC | FPMVS | EOMSC | AWMVC | FDAGF | OMVCDR | RCAGL | ALPC | TLRLF4MVC | Ours |
|---|---|---|---|---|---|---|---|---|---|---|---|---|
| | | | | | | ACC | | | | | | |
| 3sources | 0.3609 | 0.2781 | 0.3373 | 0.2959 | 0.3314 | 0.7219 | 0.6923 | 0.4320 | 0.7396 | 0.5740 | 0.3077 | **0.8343** |
| MSRC_v1 | 0.8190 | 0.8476 | 0.8000 | 0.7857 | 0.6714 | 0.8429 | 0.8286 | 0.6619 | 0.8023 | 0.8819 | 0.8857 | **0.9286** |
| BBC4 | 0.4336 | 0.2383 | 0.3591 | 0.3226 | 0.4248 | 0.6569 | **0.8832** | 0.4423 | 0.5474 | 0.7898 | 0.2788 | **0.8832** |
| BDGP_fea | 0.5244 | 0.4580 | 0.4960 | 0.4352 | 0.4208 | 0.4748 | 0.5032 | 0.5696 | 0.5036 | 0.5208 | 0.5320 | **0.6148** |
| Wiki | 0.3353 | 0.1891 | 0.3674 | 0.3143 | 0.5635 | 0.2749 | 0.4899 | 0.5335 | 0.5078 | 0.6040 | 0.1898 | **0.6110** |
| Caltech101-20 | 0.6178 | 0.4673 | 0.6605 | 0.6547 | 0.6500 | 0.4782 | 0.6186 | 0.4216 | 0.6446 | 0.3349 | 0.3630 | **0.7800** |
| Caltech101-all | 0.2912 | 0.2273 | 0.3034 | 0.2940 | 0.2232 | 0.2367 | 0.2866 | 0.1686 | 0.3206 | 0.3284 | 0.2765 | **0.4576** |
| AwA | 0.0638 | 0.0941 | 0.0434 | 0.0516 | 0.0647 | 0.0434 | 0.0919 | 0.0955 | 0.0779 | 0.0817 | 0.0994 | **0.1095** |
| | | | | | | NMI | | | | | | |
| 3sources | 0.0990 | 0.0578 | 0.1040 | 0.0769 | 0.1373 | 0.5993 | 0.5046 | 0.3087 | 0.6106 | 0.3395 | 0.0686 | **0.7252** |
| MSRC_v1 | 0.7176 | 0.7284 | 0.6992 | 0.6859 | 0.5608 | 0.7194 | 0.7150 | 0.5793 | 0.7968 | 0.7758 | **0.8986** | 0.8532 |
| BBC4 | 0.1185 | 0.0110 | 0.0686 | 0.0297 | 0.0895 | 0.4075 | **0.7111** | 0.2080 | 0.3038 | 0.5904 | 0.0122 | 0.6984 |
| BDGP_fea | 0.2642 | 0.2753 | 0.2389 | 0.2239 | 0.1459 | 0.2923 | 0.2689 | 0.3558 | 0.3038 | 0.3017 | 0.3386 | **0.3769** |
| Wiki | 0.1710 | 0.0614 | 0.2112 | 0.1715 | 0.5229 | 0.1060 | 0.3868 | 0.5389 | 0.4436 | **0.5514** | 0.0418 | 0.5127 |
| Caltech101-20 | 0.5964 | 0.6441 | 0.6420 | 0.6326 | 0.4997 | 0.6089 | **0.6789** | 0.4228 | 0.6287 | 0.0113 | 0.3815 | 0.6222 |
| Caltech101-all | 0.3542 | 0.4570 | 0.3666 | 0.3549 | 0.2470 | 0.2397 | 0.2679 | 0.3475 | 0.4158 | 0.3697 | 0.4448 | **0.4571** |
| AwA | 0.0415 | 0.1195 | 0.0523 | 0.0610 | 0.0782 | 0.0433 | 0.1012 | **0.1472** | 0.0910 | 0.0866 | 0.1181 | 0.0953 |

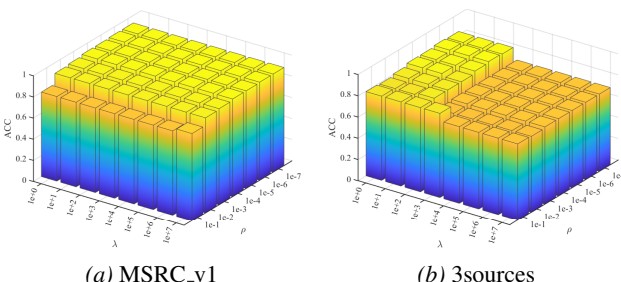

*(a)* MSRC_v1    *(b)* 3sources

*Figure 3.* Parameter Sensitivity Analysis of the Proposed Model.

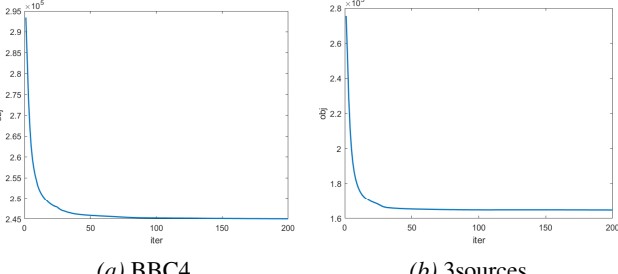

*(a)* BBC4    *(b)* 3sources

*Figure 5.* Convergence Curves of the Proposed Model on Two Datasets.

As shown in Figure 5, the convergence curves across two datasets indicate that the model loss decreases sharply in the early iterations and plateaus after about 50 iterations, effectively validating the convergence of the algorithm.

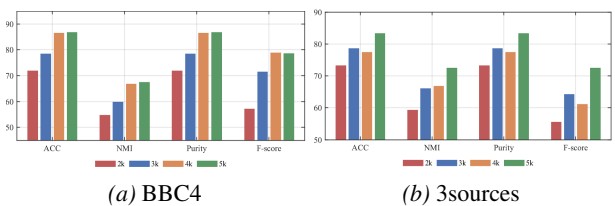

*(a)* BBC4    *(b)* 3sources

*Figure 4.* Sensitivity Analysis of the Anchor Number.

### 5.4. Ablation Study

To validate the effectiveness of each core module, we conducted systematic ablation experiments, with the results presented in Table5. Here, AVW, DF, and GP denote adaptive view weighting, discrete feedback, and graph purification, respectively. The experiments demonstrate that our

algorithm achieves the best performance, further indicating that each module contributes to the final performance. The synergistic operation of the three modules ensures the superior clustering accuracy of the complete model, thereby validating the rationality and effectiveness of our overall framework design.

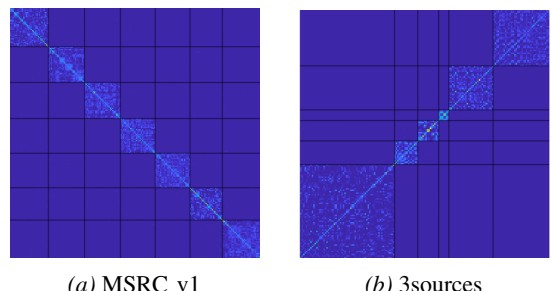

*(a)* MSRC_v1    *(b)* 3sources

*Figure 6.* Visualization of the Consensus Graph of the Proposed Model.

*Table 4.* Comparison with multi-view clustering methods.

| Datasets | SMVSC | OPMC | OMSC | FPMVS | EOMSC | AWMVC | FDAGF | OMVCDR | RCAGL | ALPC | TLRLF4MVC | Ours |
|---|---|---|---|---|---|---|---|---|---|---|---|---|
| | | | | | | Purity | | | | | | |
| 3sources | 0.4379 | 0.3964 | 0.4260 | 0.4083 | 0.4852 | 0.7751 | 0.7041 | 0.5325 | 0.7988 | 0.5976 | 0.4083 | **0.8343** |
| MSRC_v1 | 0.8190 | 0.8476 | 0.8000 | 0.7857 | 0.6714 | 0.8429 | 0.8286 | 0.6714 | 0.7816 | 0.8819 | 0.8857 | **0.9286** |
| BBC4 | 0.4584 | 0.3577 | 0.4058 | 0.3737 | 0.4482 | 0.6569 | **0.8832** | 0.5255 | 0.7810 | 0.7898 | 0.3416 | **0.8832** |
| BDGP_fea | 0.5328 | 0.4636 | 0.5108 | 0.4352 | 0.4208 | 0.4824 | 0.5272 | 0.5936 | **0.6220** | 0.5208 | 0.5884 | 0.6148 |
| Wiki | 0.3608 | 0.2261 | 0.3897 | 0.3367 | 0.5649 | 0.3011 | 0.5673 | 0.6137 | 0.5178 | **0.6399** | 0.2027 | 0.6106 |
| Caltech101-20 | 0.7062 | 0.7892 | 0.7452 | 0.7368 | 0.6542 | 0.7783 | 0.6760 | 0.5905 | **0.8504** | 0.3424 | 0.6132 | 0.7800 |
| Caltech101-all | 0.3374 | 0.4272 | 0.3480 | 0.3461 | 0.2512 | 0.2691 | 0.2619 | 0.2879 | **0.5065** | 0.3617 | 0.4182 | 0.4576 |
| AwA | **0.1386** | 0.1140 | 0.0501 | 0.0573 | 0.0659 | 0.0474 | 0.0961 | 0.1105 | 0.0801 | 0.1019 | 0.1215 | 0.1095 |
| | | | | | | F-score | | | | | | |
| 3sources | 0.2779 | 0.2161 | 0.2748 | 0.2541 | 0.2386 | **0.7365** | 0.6133 | 0.3601 | 0.6534 | 0.4867 | 0.2630 | 0.7261 |
| MSRC_v1 | 0.6988 | 0.7257 | 0.6856 | 0.6836 | 0.5475 | 0.7227 | 0.7053 | 0.6179 | 0.8000 | 0.7637 | 0.8421 | **0.8610** |
| BBC4 | 0.3385 | 0.2213 | 0.2875 | 0.2759 | 0.3647 | 0.5123 | 0.8005 | 0.2829 | 0.4854 | 0.6971 | 0.2685 | **0.8072** |
| BDGP_fea | 0.3858 | 0.3756 | 0.3714 | 0.3608 | 0.3090 | 0.3932 | 0.3905 | 0.4380 | 0.4001 | 0.4172 | 0.4383 | **0.4861** |
| Wiki | 0.2154 | 0.1359 | 0.2377 | 0.2146 | 0.4716 | 0.1770 | 0.3772 | **0.5350** | 0.4654 | 0.5105 | 0.1271 | 0.4852 |
| Caltech101-20 | 0.6679 | 0.4473 | **0.6987** | 0.6905 | 0.6110 | 0.4045 | 0.6240 | 0.4222 | 0.6735 | 0.2788 | 0.2946 | 0.3938 |
| Caltech101-all | 0.2212 | 0.2320 | 0.1903 | 0.2290 | 0.1083 | 0.0697 | 0.1669 | 0.2233 | 0.1883 | 0.2431 | **0.2927** | 0.1658 |
| AwA | **0.1045** | 0.0469 | 0.0382 | 0.0478 | 0.0510 | 0.0437 | 0.0640 | **0.0668** | 0.0543 | 0.0398 | 0.0495 | 0.0448 |

## 5.5. Qualified Study and Runtime

Figure 6 presents the final consensus similarity matrix learned by our method on two datasets. The visualization reveals a clear block-diagonal structure: prominent high-value blocks along the main diagonal, reflecting high similarity among samples of the same class, while the off-diagonal regions show uniformly low values, indicating effective separation between different classes with significantly suppressed cross-class connections. This structure visually verifies that our method can effectively overcome noise interference and learn a highly discriminative consensus graph representation.

*Table 5.* Ablation Experiments for DRMC Model.

| Datasets | | | BBC4 | | | | BDGP_fea | | | |
|---|---|---|---|---|---|---|---|---|---|---|
| AVW | DF | GP | ACC | NMI | Purity | F-score | ACC | NMI | Purity | F-score |
| | ✓ | ✓ | 0.6336 | 0.4355 | 0.6336 | 0.5166 | 0.3932 | 0.1711 | 0.3932 | 0.3150 |
| ✓ | | ✓ | 0.7679 | 0.5601 | 0.7679 | 0.6024 | 0.5104 | 0.2304 | 0.5104 | 0.3640 |
| ✓ | ✓ | | 0.8642 | 0.6678 | 0.8642 | 0.7847 | 0.5664 | 0.3108 | 0.5664 | 0.4238 |
| ✓ | ✓ | ✓ | **0.8832** | **0.6984** | **0.8832** | **0.8072** | **0.6148** | **0.3769** | **0.6148** | **0.4861** |

From the runtime comparison in Figure 7, it can be observed that the computational cost of our method on some datasets is higher than that of baseline methods. This is primarily attributed to the design of sample-level structural alignment and the discrete feedback module. While these components increase the computational burden, they enable more accurate learning of the multi-view consensus structure.

## 6. Conclusion

This paper proposes a novel framework termed Discretely-Refined Multi-view Clustering (DRMC) via Aligned Anchor

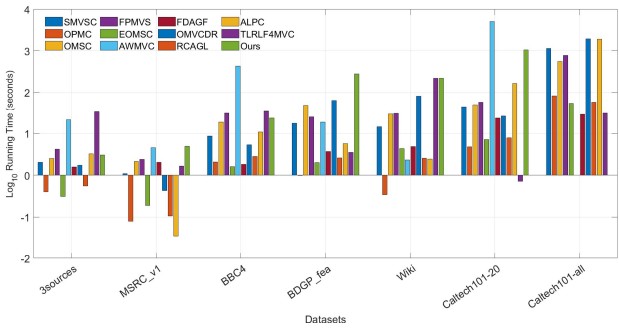

*Figure 7.* Comparative results of the execution time.

Learning. To address the limitations of the unidirectional optimization paradigm common in existing methods, DRMC constructs a sample-level consensus graph to achieve precise cross-view alignment. Furthermore, a discrete feedback module is introduced to directly incorporate clustering labels into the iterative graph purification process. This mechanism effectively bridges the gap between continuous representation learning and discrete clustering, enhancing the information exchange between them. Extensive experiments on multiple benchmarks demonstrate that DRMC outperforms state-of-the-art methods. However, the proposed method still has certain limitations. Since DRMC elevates sample-anchor relationships to sample-level similarity graphs, it introduces additional computational and storage costs compared with methods that operate only on

sample-anchor bipartite graphs. In future work, we will further improve the scalability of the method while preserving the discriminative ability of the consensus graph.

## Acknowledgments

This work was supported by the National Natural Science Foundation of China under Grants 62576067 and 62501226, the National Key Research and Development Program of China under Grant 2024YFB4710800, the Liaoning Provincial Submarine Environment Science Data Center Foundation under Grant 2025JH27/10100003, the Ministry of Industry and Information Technology of the People's Republic of China under Grant 18Q-25-06, the Liaoning Provincial Natural Science Foundation under Grants 2025-YQ-01, 2024-MS-012 and 2025-BS-0233, the Dalian Science and Technology Talent Innovation Support Plan under Grant 2024RY010, the China Postdoctoral Science Foundation under Grants 2024M760315 and 2025T180437, and the Natural Science Foundation of Hebei Province under Grant F2025201037.

## Impact Statement

This paper presents work whose goal is to advance the field of Machine Learning. There are many potential societal consequences of our work, none which we feel must be specifically highlighted here.

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
