# OpenReview forum: "Discretely-Refined Multi-view Clustering via Aligned Anchor Learning"
_ICML.cc/2026/Conference — ICML 2026 regular_

### Official Review · Reviewer_xRZe · 2026-02-28

**Soundness:** 3
**Presentation:** 4
**Significance:** 4
**Originality:** 3
**Overall Recommendation:** 5
**Confidence:** 5

**Summary:**

This paper proposes a discretely-refined multi-view clustering method based on anchor learning. The method elevates anchor graphs into sample-level similarity and introduces a bidirectional feedback module to dynamically optimize the consensus graph using the obtained discrete labels, aiming to address the limitations of existing methods in terms of insufficient view alignment and unidirectional optimization from graph structure to cluster labels.

**Compliance With Llm Reviewing Policy:**

Affirmed.

**Final Justification:**

All the concerns have been stressed. I will keep my positive score.

**Key Questions For Authors:**

(1) Does the algorithm involve any random initialization? If so, how were the random seeds set?
(2) The experimental results show that the proposed method achieves significant improvements on the ACC metric, while the improvements on Purity and F-score are relatively smaller. Please analyze the possible reasons for this phenomenon.
(3) In the ablation study, when each of the three modules is removed individually, were the parameter settings of the other modules kept consistent with those of the full model? Was parameter tuning performed again for each variant?

**Limitations:**

Yes

**Strengths And Weaknesses:**

This paper has the following strengths:
(1) In response to the limitations of existing methods, which only perform alignment at the anchor graph level and suffer from unidirectional optimization between graph learning and label assignment, this paper proposes a targeted introduction of sample-level similarity alignment and a bidirectional feedback mechanism.
(2) The methodology is presented comprehensively, and experimental results demonstrate that the proposed method achieves strong competitiveness on multiple datasets, with particularly significant improvements on the ACC metric.
(3) The ablation study is well-designed and rigorously validates the contribution of each module to the final performance.
But I also have some suggestions:
(1) The list of references could be expanded. Including more representative works in the field would help readers gain a more comprehensive understanding of the research progress and background.
(2) The conclusion section only summarizes the main contributions and performance of the proposed method. A discussion of its limitations would make the conclusion more comprehensive.
(3) The abbreviations used to describe each module in the ablation study are inconsistent with those labeled in Table 4. It is recommended to unify the notation to avoid confusion.

---

> ### Author Rebuttal · Authors · 2026-03-30
>
> Thank you for your detailed reading and advice.
>
> **Q1: Does the algorithm involve any random initialization?**
>
> A1: The method in this paper adopts deterministic initialization rather than random initialization, where the initial values of all variables are given by fixed initialization rules. This helps reduce performance fluctuations caused by random initialization and improves the stability of the experiments.
>
> **Q2: Why are the improvements on Purity and F-score are relatively smaller?**
>
> A2: The difference in results may stem from the optimization focus of the DRMC method. The graph purification module in DRMC places greater emphasis on inter-class separability and the correctness of main cluster assignment, leading to a significant improvement in ACC. In contrast, F-score is more sensitive to local boundary samples and samples with heterogeneous cluster compositions, resulting in a smaller improvement.
>
> **Q3: In the ablation study, was parameter tuning performed again for each variant?**
>
> A3: In the ablation experiments, we held the same parameter settings as the full model, only removing the corresponding modules without retuning the hyperparameters for the ablated variants. This allows for a fairer assessment of the contribution of each individual module, free from the influence of other factors.
>
> **Q4: The issues regarding form and details.**
>
> A4: We will further expand the reference list by adding representative works in related directions from recent years, to help readers gain a more comprehensive understanding of the research progress and background in the field. We will also add a discussion on the limitations of the method, enabling readers to better understand its applicable scenarios and directions for future improvement.

---

> > ### Author Rebuttal · Reviewer_xRZe · 2026-04-01
> >
> > All the concerns have been stressed. I will keep my positive score.

---

### Official Review · Reviewer_SRnR · 2026-03-10

**Soundness:** 3
**Presentation:** 3
**Significance:** 3
**Originality:** 4
**Overall Recommendation:** 5
**Confidence:** 4

**Summary:**

This paper proposes a novel anchor-based multi-view clustering method named DRMC, whose main innovations lie in the design of a sample-level view alignment module and a bidirectional feedback module, effectively enhancing the discriminability of the consensus graph. Extensive experiments on multiple benchmark datasets demonstrate the competitive performance of the proposed method.

**Compliance With Llm Reviewing Policy:**

Affirmed.

**Key Questions For Authors:**

1. Regarding the baseline methods, were the reported results obtained with the parameters tuned within the ranges recommended by their respective authors?
2. Algorithm 1 lists multiple variables that need to be initialized. How were these variables specifically initialized?
3. The paper only provides the value ranges for the hyper-parameters but does not specify the step size. How was the step size set during parameter search?

**Limitations:**

yes

**Strengths And Weaknesses:**

Strengths:
1. This paper clearly identifies the limitations of existing methods in terms of the view alignment level and the unidirectional optimization from graph structure to cluster labels, and accordingly proposes targeted solutions to address these issues.
2. The methodology is presented with clarity, and the proposed framework intuitively illustrates the core innovations.
3. The experimental content is relatively comprehensive. By comparing with various existing methods on multiple benchmark datasets, the effectiveness of the proposed method is validated.
Weaknesses:
1. The form of the sparse regularization term defined in the objective function differs slightly from that adopted in the optimization process. It would be helpful to further clarify the transformation relationship between the two.
2. In the methodology section, the notation for the spectral embedding matrix F and the discrete label matrix Y is inconsistent, with superscripts appearing in some instances. It is recommended to unify them throughout the paper to improve readability. In addition, the terminology used for R in the optimization subsection does not fully align with its description in the notation table.
3. It would be beneficial to provide more details in the experimental setup regarding the fairness of the comparison methods, such as whether the parameters were tuned within their respective optimal ranges.

---

> ### Author Rebuttal · Authors · 2026-03-30
>
> Thank you for your careful reading and valuable suggestions.
>
> **Q1: Were the method parameters of the baseline results adjusted within the specified range?**
>
> A1: The experimental results of the compared methods are either based on their publicly available implementations or reproduced following the configurations recommended in the original papers. On this basis, we further performed a grid search within the parameter ranges suggested in the respective papers, and ultimately report the best achievable results for each method to ensure fair comparison.
>
> **Q2: How were the variables specifically initialized?**
>
> A2: In Algorithm 1, the initial view weights are set to a uniform distribution to ensure that each view contributes equally at the initial stage. The consensus graph is initialized as a zero matrix, and the rotation matrix is initialized as the identity matrix. The initial spectral embedding matrix is obtained by performing eigen-decomposition on the Laplacian of the current graph, and the initial discrete label matrix is further obtained through a discrete consistency step.
>
> **Q3: How was the step size set during parameter search?**
>
> A3: The hyperparameters differ by an order of magnitude; that is, the specific hyperparameter values are taken as:
> $\lambda \in \lbrace 1e^0, 1e^1, 1e^2, 1e^3, 1e^4, 1e^5, 1e^6, 1e^7\rbrace$，$\rho \in \lbrace 1e^{-1}, 1e^{-2}, 1e^{-3}, 1e^{-4}, 1e^{-5}, 1e^{-6}, 1e^{-7}\rbrace$. We will add this clarification later to make the hyperparameter settings clearer.
>
> **Q4: The formal issues.**
>
> A4: The second term in Eq. (1) represents an ideal sparse prior, aiming to associate each sample with only a few anchors. However, the $\ell_0$ term is non-differentiable and difficult to optimize directly. Therefore, in Eq. (5) and the actual optimization, we adopt a differentiable surrogate form to facilitate subsequent optimization. Meanwhile, we will carefully review the entire manuscript to unify the notation and ensure consistency throughout the paper.

---

### Official Review · Reviewer_qE4a · 2026-03-12

**Soundness:** 4
**Presentation:** 4
**Significance:** 3
**Originality:** 4
**Overall Recommendation:** 4
**Confidence:** 4

**Summary:**

The main contributions of this paper include: (1) designing a sample-level view alignment and fusion mechanism to align cross-view information at the sample level; (2) constructing a bidirectional feedback module to form mutual optimization between the graph structure and label assignment; (3) adopting an adaptive weighted fusion strategy to learn a more discriminative consensus graph.

**Compliance With Llm Reviewing Policy:**

Affirmed.

**Final Justification:**

The authors addressed my main problems and concerns.

**Key Questions For Authors:**

(1) Can the proposed method maintain stable clustering performance when the sample size is very small?
(2) Please explain the purpose of imposing the orthogonal constraint on the rotation matrix. What impact would the absence of this constraint have on the stability of the final cluster indicators?
(3) How should "Discretely-Refined" in the title be understood, and what is the essential difference between this concept and the traditional approach of first obtaining continuous spectral embeddings followed by post-processing discretization?

**Limitations:**

Yes

**Strengths And Weaknesses:**

Strengths:
(1) Previous methods typically follow a unidirectional process from spectral embedding to discrete labels, where clustering results cannot be fed back to refine the graph structure. The bidirectional feedback module proposed in this paper enables mutual correction between the graph structure and clustering results, which is relatively novel.
(2) By introducing a rotation matrix to establish correspondence between spectral embeddings and discrete labels, the proposed method avoids the information loss caused by the traditional post-processing pipeline of first performing spectral clustering followed by k-means.

Weaknesses:
(1) The analysis of the ablation study results could be more detailed. Which module contributes more to the overall performance, and what are the underlying reasons?
(2) In Eq. (19), the authors impose an orthogonal constraint on the rotation matrix R, but provide insufficient explanation regarding the role of this constraint, which is essential for understanding the model.

---

> ### Author Rebuttal · Authors · 2026-03-30
>
> We are grateful for your great praise and valuable comments on our work.
>
> **Q1: Can the proposed method maintain stable clustering performance when the sample size is very small?**
>
> A1: The proposed method maintains a certain degree of stability in small-sample scenarios. Although the structural information between samples becomes sparser when the sample size is small, which may lead to certain performance fluctuations, the core of the method relies on leveraging multi-view structural consistency to suppress noise interference and enhancing the discriminability of the consensus graph through a feedback mechanism. These designs are not strictly depend on large-scale samples.
>
> **Q2: Explain the purpose of imposing the orthogonal constraint on the rotation matrix.**
>
> A2: The purpose of imposing the orthogonal constraint on the rotation matrix is as described in the second point above. Removing this constraint would make the correspondence between the continuous embedding $F$ and the discrete labels $Y$ unstable, thereby reducing the stability of the final clustering metrics and making the results more prone to fluctuations.
>
> **Q3: How should "Discretely-Refined" in the title be understood?**
>
> A3: Unlike existing spectral clustering methods, the “Discretely-Refined” in DRMC is not merely a post-processing step performed after obtaining the continuous spectral embedding. Instead, it introduces discrete clustering information into the iterative optimization process to continuously purify the consensus graph. Discrete labels serve not only as the final output but also as feedback that guides the refinement of the graph structure, thereby forming an iterative optimization loop between graph learning and label assignment. This design enables the continuous representation and discrete labels to mutually reinforce each other, enhancing the discriminability of the consensus graph and improving clustering performance.

---

> > ### Author Rebuttal · Reviewer_qE4a · 2026-04-01
> >
> > My problems have been solved, I have no further questions.

---

### Official Review · Reviewer_Ep94 · 2026-03-16

**Soundness:** 2
**Presentation:** 2
**Significance:** 2
**Originality:** 2
**Overall Recommendation:** 3
**Confidence:** 4

**Summary:**

This paper proposes DRMC for anchor-based multi-view clustering. It lifts anchor-sample bipartite graphs to full n×n sample-level similarity matrices for cross-view alignment, fuses them via adaptive inverse-distance weighting, and introduces a bidirectional feedback loop where discrete cluster labels that obtained via orthogonal Procrustes alignment purify the consensus graph iteratively. Experiments on eight datasets against twelve baselines show strong ACC on most datasets.

**Compliance With Llm Reviewing Policy:**

Affirmed.

**Key Questions For Authors:**

1. Can you provide a convergence guarantee that accounts for the graph purification step, or at least empirical evidence that the objective decreases monotonically?
2. Why does F-score/NMI underperform on several datasets despite strong ACC? Is the feedback loop biased toward pure but imbalanced clusters?
3. What is the wall-clock time for n > 50K? This would clarify whether the O(n²) cost is acceptable in practice.

**Strengths And Weaknesses:**

Strengths
The bidirectional feedback between discrete labels and graph structure is a well-motivated contribution that addresses the limitations of existing unidirectional AMVC pipelines. The framework figure (Figure 2) clearly presents the approach. Most sub-problems admit closed-form or efficient updates. The comparison spans twelve baselines across eight datasets with four metrics, and the ablation study systematically evaluates each module.
Weaknesses
1. Scalability contradicts motivation. Constructing full n×n similarity matrices gives O(n²) time and space complexity, defeating the purpose of anchor-based methods designed for large-scale efficiency. The largest dataset tested has n=30K. No discussion of practical limits or comparison with near-linear AMVC methods on scalability.
2. No convergence guarantee. No monotonic decrease argument or convergence proof is provided. Empirical convergence curves alone are insufficient.
3. Heuristic graph purification. The adaptive γ with clipped bounds [0.01, 0.5] lacks justification. Feeding unreliable early-stage labels back into the graph risks error amplification. This is not analyzed.
5. Presentation issues. Table 4 duplicates Table 3’s caption. Notation overloads v for both view index and total views. Complexity analysis appears before the optimization section.

---

> ### Author Rebuttal · Authors · 2026-03-31
>
> Thank you for your valuable feedback on our work.
>
> **Q1: Can provide a convergence guarantee that accounts for the graph purification step?**
>
> A1: It should be noted that the final function of the method is not exactly the same as the optimization function. Considering the solvability of the optimization process, we introduced an entropy regularization term for the view weights $\alpha_v$ and a graph purification step. Therefore, from the optimization perspective, the overall optimization function consists of the original objective function, the entropy regularization term, and the graph purification step. Under this view, the whole algorithm can be regarded as an alternating optimization procedure. In each iteration, the variables are updated sequentially while fixing the others, so the objective value is non-increasing under the corresponding subproblem updates. In addition, all squared terms are nonnegative, the graph Laplacian term is lower-bounded due to the positive semidefiniteness of the Laplacian matrix, and the entropy regularization term is also lower-bounded under the constraints $\alpha_v > 0$ and $\sum_{v=1}^{V} \alpha_v = 1$. Therefore, the objective-value sequence is monotonically non-increasing and lower-bounded, which provides a convergence interpretation for the optimization process. We will further clarify this unified view and the corresponding analysis in the next version.
>
> $$
> J = \underbrace{\sum\_{v=1}^{V} \||X^{(v)} - A^{(v)}U^{(v)}\||\_{F}^2 + \rho \sum\_{v=1}^{V} \Omega(A^{(v)}) + \lambda \left(\sum\_{v=1}^{V} \alpha\_v \||W^{(v)} - \overline{W}\||\_{F}^2 + \text{tr}(F^T L(\overline{W})F) + \||F - YR\||\_F^2\right)
> }\_{\text{original objective}}+\underbrace{ \lambda \left( - \tau \sum_{v=1}^{V} \log \alpha\_v \right)}_{\text{entropy regularization term}} + \underbrace{ \mu \|| (11^T - S) \odot \widetilde{W} \||\_F^2 }\_{\text{graph purification}}
> $$
>
> **Q2: Why does F-score/NMI underperform on several datasets?**
>
> A2: The above phenomenon, in our view, is mainly related to the different emphases of the evaluation metrics. Specifically, ACC focuses on the overall alignment between the clustering results and the ground-truth labels, whereas NMI and F-score are more sensitive to the consistency of class distribution and pairwise sample relationships. From the perspective of the proposed mechanism, the feedback loop in our method is mainly designed to enhance the discriminability of the consensus graph and suppress connections between cross-cluster samples, rather than encouraging cluster-size imbalance. Therefore, for issues such as class imbalance and large view-quality variations that exist in some datasets, we do not believe they should be attributed to the feedback mechanism biasing the model toward producing pure but imbalanced clustering results.
>
> **Q3: The issue of time complexity.**
>
> A3: Compared with linear anchor graph methods, the advantage of DRMC lies in that it does not stop at anchor-level relations, but further lifts them to sample-wise similarity relationships. This makes the method less sensitive to the quality of the initial anchors and helps achieve better clustering performance. Moreover, compared with methods of similar complexity, our approach also obtains more competitive results. The proposed DRMC requires more computational time when handling large-scale datasets, such as those with $n > 50k$, due to its $O(n^2)$ time complexity. However, it should be clearly emphasized that the actual wall-clock time depends on factors such as hardware configuration and the degree of parallelization. Therefore, with high-performance devices or parallel implementations, the time overhead can still be further reduced. We will further clarify the applicable scenarios of our method and discuss its complexity limitations in the revised version.
>
> **Q4: The adaptive γ with clipped bounds [0.01, 0.5] lacks justification.**
>
> A4: The goal of graph purification is to refine the current consensus graph rather than reconstruct it from scratch. In this process, $\gamma$ serves as a purification coefficient that controls the suppression strength on unreliable inter-cluster edges. A small $\gamma$ weakens cross-cluster connections while preserving the main structural information in the original graph. Accordingly, $\gamma$ is restricted to $[0.01, 0.5]$. If $\gamma \to 1$, the suppression effect becomes negligible; if $\gamma > 1$, inter-cluster edges are amplified, which is contrary to the purpose of purification and usually degrades performance. The lower bound $0.01$ is used to maintain numerical stability. We will further clarify the role of $\gamma$ and the rationale for its range in the next version.

---

### Decision · Program_Chairs · 2026-04-30

**Decision:**

Accept (regular)

**Comment:**

Overall, reviewers showed a high level of agreement on the main strengths of this paper, particularly its technical novelty, sound methodological design, and strong empirical performance. In rebuttal, authors also carefully and effectively responded to the major concerns of  reviewers, especially those related to the optimization process (such as convergence), experimental analysis, and computational cost. Several reviewers explicitly stated that their questions had been fully addressed and therefore maintained their positive evaluations. While a few issues still deserve further clarification in the revision stage, they do not fundamentally weaken the novelty, technical soundness, or practical value of the proposed method. Taken together, the rebuttal record is clearly positive, and I believe this paper is worthy of acceptance.